# Explaining the flaws in human random generation as local sampling with momentum

**Lucas Castillo**[1]*, **Pablo León-Villagrá**[2], **Nick Chater**[3], **Adam Sanborn**[1]

**1** Department of Psychology, University of Warwick, Coventry, United Kingdom, **2** Cognitive, Linguistic & Psychological Sciences, Brown University, Providence, Rhode Island, United States of America, **3** Warwick Business School, University of Warwick, Coventry, United Kingdom

* castillo.lucas@protonmail.com

## Abstract

In many tasks, human behavior is far noisier than is optimal. Yet when asked to behave randomly, people are typically too predictable. We argue that these apparently contrasting observations have the same origin: the operation of a general-purpose local sampling algorithm for probabilistic inference. This account makes distinctive predictions regarding random sequence generation, not predicted by previous accounts—which suggests that randomness is produced by inhibition of habitual behavior, striving for unpredictability. We verify these predictions in two experiments: people show the same deviations from randomness when randomly generating from non-uniform or recently-learned distributions. In addition, our data show a novel signature behavior, that people's sequences have too few changes of trajectory, which argues against the specific local sampling algorithms that have been proposed in past work with other tasks. Using computational modeling, we show that local sampling where direction is maintained across trials best explains our data, which suggests it may be used in other tasks too. While local sampling has previously explained why people are unpredictable in standard cognitive tasks, here it also explains why human random sequences are not unpredictable enough.

## Author summary

When explicitly asked to be random, people are not random enough. Previous accounts of these random generation tasks have argued that people are effortfully trying not to be predictable. In many other tasks, however, people also show random behavior, even when it is unnecessary or outright disadvantageous. Here, we try to bridge this apparent gap. We hypothesize that the randomness people produce when trying to be random and the randomness that they display when trying to make the best choice has the same common mechanism: drawing mental samples to make judgments and decisions. In two experiments, we compare previous random generation accounts, which are task-specific in nature, to the more general account of mental sampling that has been used to explain how people behave in many other domains. We find that the flexibility of human random generation in our data is better explained by the mental sampling account. We also find a novel empirical signature of momentum in random generation, which points to a new

**Data Availability Statement:** All data and computational code are available on OSF at https://osf.io/dw8ez/.

**Funding:** LC, PLV and ANS were funded by a European Research Council (https://erc.europa.eu/)

Consolidator grant (817492-SAMPLING) awarded to ANS. The funders had no role in study design, data collection and analysis, decision to publish, or preparation of the manuscript.

**Competing interests:** The authors have declared that no competing interests exist.

kind of mental sampling algorithm. If mental sampling governs behavior in random generation tasks and elsewhere, then this task has great promise in helping to understand wider human behavior.

## Introduction

In many tasks, people behave with some degree of randomness, even when they are not required to do so. For example, in a task involving several repeated gambles, participants might make a different choice when presented with the same set of options for a second time [1]. Surprisingly, people behave randomly even to their disadvantage: when given options with different reward probabilities, participants will choose each alternative proportionally to the probability of being rewarded, rather than always choose the most advantageous option [2, 3].

To account for these inconsistencies in people's behavior, almost all models of cognition postulate that mental mechanisms include sources of randomness, typically modeled as independent, identically distributed (*iid*) samples. This source of randomness within cognition is used to explain the noisiness of behavior, whether in higher-level cognitive processes such as categorization or decision-making [4–7], and in lower-level processes such as perception [8, 9].

Whether people can produce randomness has also been studied more directly, by asking participants to generate sequences of items unpredictably. Having the ability to behave randomly is important in adversarial situations, where being unpredictable is the optimal behavior [10, 11], and perceiving someone as unpredictable is seen as an indicator of free will [12, 13]. In addition, departures from randomness have helped model the cognitive architectures of neurotypical and neurodivergent populations [14, 15].

In a typical random generation experiment, participants are given a set of items (usually numbers from 1 to 10) and are asked to produce them unpredictably, which instructions will often exemplify as drawing items 'out of a hat' with replacement. Variations of the task have involved doing the task vocally or using a keyboard or mouse [16], performing the task at different speeds [17], while multi-tasking [18], or collaboratively [19].

Paradoxically, despite people's tendency to behave randomly in a myriad of domains that do not require them to, this body of work has found that when asked to be unpredictable, people are not random enough. Across experiments, the same picture emerges: people's sequences are typically more compressible than truly random sequences [15] (c.f. [20]) and display predictable patterns of serial dependence [21, 22], thus deviating significantly from the *iid* sampling that many cognitive models include.

Previous accounts of people's behavior in random generation tasks make no connection between people's excessively unpredictable behavior in many perceptual and cognitive tasks from their performance when explicitly generating random sequences. Instead, they characterize being random as the product of effortful behavior. For example, in their *network modulation model* Jahanshahi et al. [23] theorize that in a random generation task people create an associative network with each possible response as a node, and with links (representing the probability of transitioning from one item to another) having weights proportional to the items' strength of association. If operating alone, this network would produce greatly stereotyped responses, but a second component of their model—the controller—inhibits the strongest links to enable variability, while monitoring the output to modulate its intervention on the network. Similarly, another popular account, Baddeley's *schema account* [24], postulates that in random generation tasks people follow deterministic, learned action sets (schemas),

and switch between sets based on a monitoring process that evaluates how unpredictable the resulting sequence is and changes strategy if randomness is perceived to decline.

According to these accounts, people's deviations from randomness can be explained by biases regarding which schemas are preferred and limitations in how often schemas are changed and how well randomness is monitored; or by limitations in the controller's ability to inhibit links in the associative network. While these approaches can explain people's flaws when producing random sequences, they are, however, specific to the requirements of the random generation task, and would not apply to other tasks in which randomness has been observed.

Recent work has, however, raised the possibility that a single mechanism—local sampling—might explain both the excessive noisiness of many aspects of human behavior and the excessive predictability of random sequence generation. It turns out that randomness in other tasks is also typically not *iid* [25]. Instead, the noise in people's behavior has a rich structure, including long-term autocorrelations [26, 27]. Moreover, it has been suggested that these patterns arise from a general-purpose approximation to probabilistic inference [28] widely used in statistics and machine learning. These *local sampling* algorithms generate new samples from the previous one, creating sequential dependencies [29]. Local sampling algorithms have been used to explain how people reason with probabilities [28], including the characteristic judgment errors people make [30, 31], and have also been proposed in causal learning [32], bistable perception [8], memory retrieval [33], and elsewhere [34].

Here, we postulate that local sampling underpins the attempt to generate sequences. This would have implications for the domains in which random generation has been employed, that is, for how we understand people's behavior in adversarial situations, free will, and neurodiverse populations. In addition, random generation tasks could reveal undiscovered aspects of the underlying local sampling mechanism, shedding light on how people perform a wide range of tasks involving probabilistic inference.

Previous accounts of random generation have been constructed to deal with cases where people must choose from a uniform distribution over a single dimension (e.g. numbers), often with ordered items, and expect that people will draw samples uniformly: the network modulation model achieves unpredictability by attempting to make each possible bigram equally likely, while the schema model does so by changing which schemas are used based on unpredictability of the sequence alone. Crucially, the local sampling account is much more general, predicting that people will be able to draw samples from any distribution while matching their probabilities, including non-uniform distributions or multivariate distributions. This leads to a crucial differential prediction between our proposal and previous models, which center on uniform distributions and which postulate that people strive for unpredictability only. In contrast, local sampling accounts assume that sampling will match the underlying distribution density learned by the participant.

A second differential prediction is that, according to the schema account and the network modulation model, many of the patterns that arise from human random generation do so from the existence of habitual behaviors that must be inhibited. In case of the network modulation model, the associations between items have different strengths, and the controller evens these out to increase variability. In case of the schema account, transitions between items are due to the application of well-learned transformations (schemas). These accounts have been applied to tasks where the items to randomize were known beforehand, and so it is unclear from these accounts how participants would perform when trying to randomize sets that have recently been learned.

Here, we devised two novel random generation tasks exploring whether people can generate non-uniformly-distributed items, or items they have recently been learned. Importantly,

we wanted to see whether people generated items in these novel tasks while still displaying the typical departures from randomness that are often seen in the typical random generation paradigm, where items are well-known and uniformly distributed, as that would point to the same cognitive mechanism being used throughout. In Experiment 1, we asked participants to produce a sequence of random heights, and tested whether their sequences reflected the true, approximately Gaussian, distribution of heights. In Experiment 2, we taught participants a set of items configured in either a one-dimensional or a two-dimensional display, and tested whether they could generate random items in these domains.

We found that people could generate random items in these new tasks, and that their sequences exhibited the same systematic deviations from randomness found elsewhere in the random generation literature [35], pointing to a common mechanism across all these tasks. Crucially, as the local sampling account predicts, participants were sensitive to the distributional properties of the domain, being able to reproduce non-uniform distributions in their samples. Finally, we observed a key systematic deviation from *iid* sampling—that people follow the same trajectory for multiple trials over and above what would be expected from their making small transitions only.

We computationally modeled these qualitative observations, identifying which forms of local sampling explain human data best and contrasting them with Cooper's [36] schema model. Because no computational model is available for the network modulation account, we do not include it in our model comparisons, but return to it in our discussion. We found that data were most closely matched by a local sampling algorithm with "recycled momentum", an algorithm which has not previously been suggested to underlie human sampling. Our analysis also shows that random generation tasks are useful for identifying subtle differences between different candidate algorithms, opening the door to more such experiments in future.

## Results

In Experiment 1, we tested whether participants could sample random items non-uniformly while displaying the same deviations from *iid*. Participants produced a random sequence of heights of either men or women in the United Kingdom. In one sequence, they sampled heights as distributed according to a uniform distribution (Uniform condition); in the other sequence, heights were distributed following their actual distribution (which is roughly Gaussian [37], and so we term this the Gaussian condition). The order in which participants produced these sequences was counterbalanced. In Experiment 2, we tested whether participants could sample from a set of novel items while displaying the same deviations from *iid*. Participants first learned a set of syllables arranged in either a single row (one-dimensional condition) or a grid (two-dimensional condition; see Fig 1B), then produced two random sequences for the same display.

### Deviations from *iid* sampling

To evaluate whether people deviated from *iid* sampling in the same way as in previous random generation experiments, we evaluated several properties of the sequences commonly investigated in past research [21] and whether these deviated from the properties expected from a random sequence [38] (see Fig 1 for examples of the first three measures). We focused on deviations from serial independence, as they can easily be applied to non-uniform distributions and are more interpretable than compressibility measures, which can in turn inform building better cognitive models. The indices were:

- *Repetitions*: The proportion of transitions where the new item was a repetition of the last.

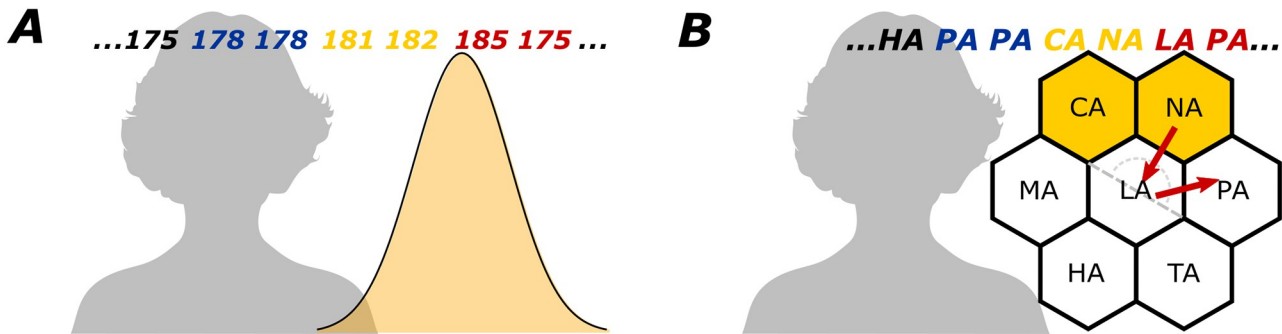

**Fig 1. Example measures.** Examples of a repetition (blue), an adjacency (yellow) and a turning point (red) in (A) a sequence of normally-distributed heights (in cm) in the Gaussian condition of Experiment 1, and (B) a sequence of syllables whose arrangement had recently been learned, for the two-dimensional condition of Experiment 2. Woman's outline image source: https://commons.wikimedia.org/wiki/File:Black_-_replace_this_image_female.svg.

- *Adjacencies*: The proportion of transitions where the new item was one unit distance away to the one prior. To ensure that this analysis reflected people's transition patterns, we analyzed this measure after removing all repeated items from the sequence. By applying this correction, we avoided having values for this measure depend on how often repetitions occur (for example, without this correction, *Repetitions* and *Adjacencies* would be negatively correlated).

- *Turning Points*: The proportion of transitions that did not follow the previous direction. In previous random generation experiments using the number line, this has been defined as a transition that begins a descending run after having followed an ascending run (e.g. "1, 4, 2"), or vice versa. For Experiment 2, we generalize this measure to describe turning points in the spatial displays: we define it as a transition for which the absolute difference between the current and previous direction is larger than 90 degrees (see Fig 1). Again, we analyzed this measure after removing all repeated items from the sequence.

- *Distances*: The average Euclidean distance traveled in each transition. We analyzed this measure after removing all repeated items from the sequence.

In previous random generation literature, people generating numbers or letters have been found to deviate from *iid* sampling in that they repeat items infrequently and transition between items making small jumps and following the same trajectory for multiple utterances. For this reason, if the same mechanism was used to generate items in these novel tasks, we would expect higher *Adjacencies* and lower *Distances*, lower *Repetitions*, and lower *Turning Points*, than *iid* sampling.

To compare participant's values to those that would be expected from *iid* sampling, we reshuffled each participant's sequence $10^4$ times and obtained the average value of each index across reshuffled sequences. If an index required removal of repetitions, we first reshuffled the original sequence with all items, and then removed items that were a repetition of the last in the new reordering. We ran generalized linear mixed-effects models predicting the observed values, using a logit link function for the first three measures and the identity link function for *Distances* (see Eq 1). We included the *iid* expectation as an offset variable (coefficient set to 1), so that the value of $\beta_0$ represented the difference between observed and expected values. We also included a random intercept per participant ($u_i$).

$$ObservedValue = \beta_0 + 1 \times ExpectedValue + u_i \tag{1}$$

We also compared these results to a model that included a regressor on experimental condition (Eq 2), which allowed us to examine whether potential deviations from *iid* sampling depended on the domain participants were producing from ($\beta_1$).

$$ObservedValue = \beta_0 + \beta_1 \times Condition + 1 \times iidValue + u_i \qquad (2)$$

We only report condition differences where the conditions displayed different qualitative trends, relegating other analyses to S2 Text.

Finally, because each participant produced two sequences, we added *Order* and *Order × Condition* terms to the model above to ensure that the above results did not depend on whether the sequence was their first or their second. We found no qualitative differences due to either term, and so we relegate the report on those analyses to S3 Text.

**Experiment 1.** We found that participants deviated from their reshuffled sequences in the same systematic way as in previous random generation experiments. Compared to their reshuffled sequences, participants' values were lower for *Repetitions* (Observed = .015, Expected = .043, $Z = -2.61$, $p = .009$, $d = -0.67$, $BF_{10} = 4$), *Turning Points* (Obs. = .47, Exp. = .65, $Z = -11.12$, $p < .001$, $d = -0.64$, $BF_{10} = 4.2 \times 10^6$) and *Distances* (Obs. = 9.98, Exp. = 18.42, $t(18.98) = -3.36$, $p = .003$, $d = -0.42$, $BF_{10} = 7$), and higher for *Adjacencies* (Obs. = .22, Exp. = .07, $Z = 6.15$, $p < .001$, $d = 0.97$, $BF_{10} = 2.6 \times 10^3$).

**Experiment 2.** Likewise, in Experiment 2, participants deviated from their reshuffled sequences in the same systematic way as in previous random generation experiments. Both in the one-dimensional and two-dimensional conditions, participants had lower *Repetitions* (Obs. = .04, Exp. = .15, $Z = -9.55$, $p < .001$, $d = -1.31$, $BF_{10} = 4.6 \times 10^7$) and lower *Turning Points* (Obs. = .66, Exp. = 0.70, $Z = -3.70$; $p < .001$, $d = -0.17$, $BF_{10} = 5$). In the one-dimensional condition, they also had lower *Distances* (Obs. = 2.22, Exp. = 2.70, $t(19.01) = -6.58$, $p < .001$, $d = -0.32$, $BF_{10} = 703$) and higher *Adjacencies* (Obs. = .46, Exp. = .29, $Z = 6.27$, $p < .001$, $d = 0.61$, $BF_{10} = 340$) than *iid*, but these did not differ in the two-dimensional condition (Obs. = 1.47, Exp. = 1.37, $t(18.99) = 1.60$, $p = .13$, $d = 0.13$, $BF_{10} = 1/8$; and Obs. = .58, Exp. = .55, $Z = 1.61$, $p = .11$, $d = 0.08$, $BF_{10} = 1/12$; respectively).

## Distributional sensitivity

A key distinction between the predictions of the schema and local sampling accounts is whether people can generate random sequences that do not follow a uniform distribution. Comparing aggregate and individual distributions corresponding to the random sequences, we found that, in aggregate, Uniform and Gaussian conditions produced different item distributions (see Fig 2A). This difference was not merely due to data aggregation, since most of the individual participants also produced different distributions in the two conditions (see Fig 2C). In order to quantitatively study whether participants' sequences resembled a uniform or a Gaussian distribution more, we developed a new measure, which we term *Shape*, which was calculated as:

$$S = \frac{1}{N} \sum_{n=1}^{N} lpdf_G(x_n) - lpdf_U(x_n)$$

where $lpdf_G(\cdot)$ and $lpdf_U(\cdot)$ are the log densities of the best fitting Gaussian and uniform distribution respectively, $x_n$ is each item in the sequence, and $N$ is the total length of the sequence. *Shape* values are positive when a sequence is better described by a Gaussian distribution rather than a uniform distribution, and vice versa. We did not remove repeated items before calculating this measure. Initially, we pre-registered that we would use a different measure: we would fit sequences to a normal and a uniform distribution, and use BIC values to classify participants' sequences. However, we decided against using this measure in favor of the *Shape* measure.

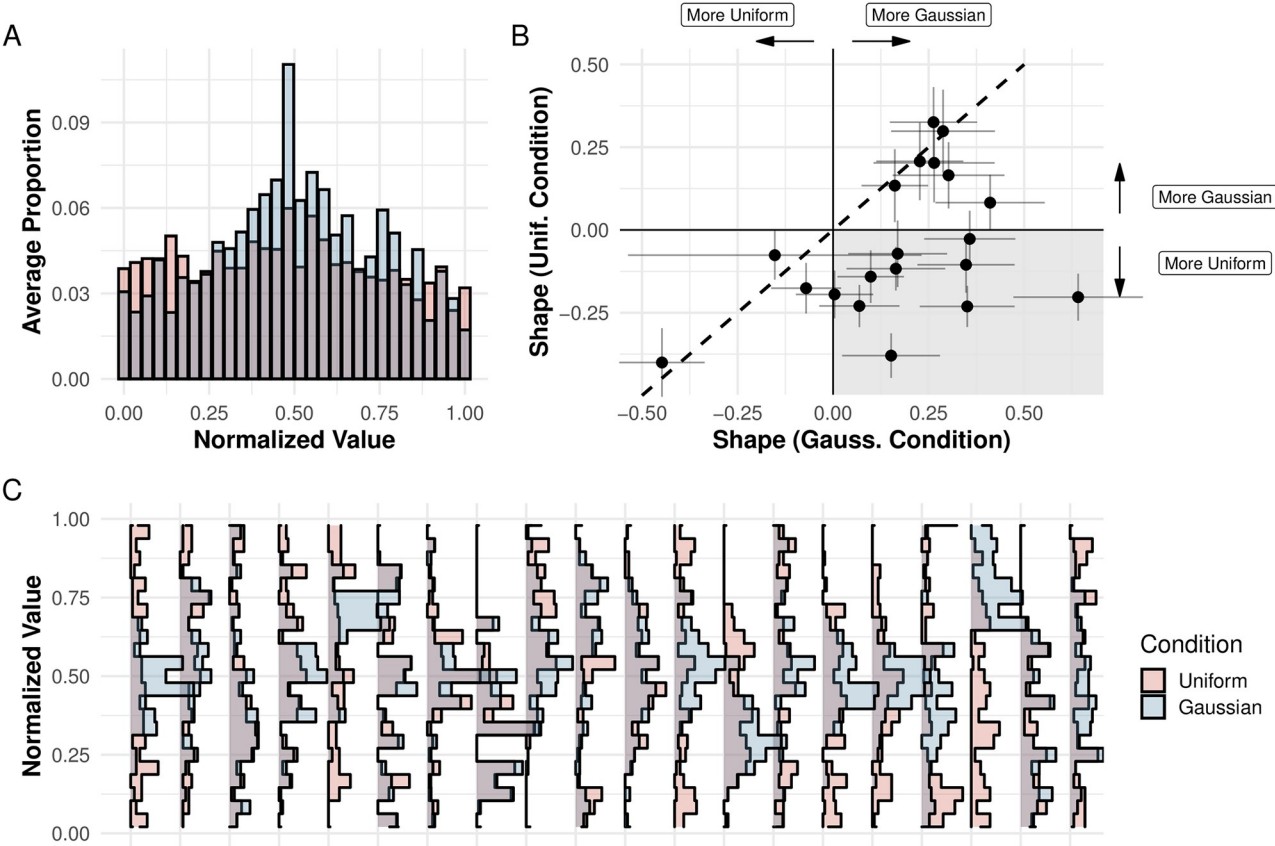

**Fig 2. Distributional Sensitivity Results.** Distribution of items for participants in the Gaussian and Uniform conditions in Experiment 1. To ease visualization, we normalized each participant's values. (A) We calculated the average proportion of values of each participant in each of thirty bins, so that each participant had equal weight in the resulting plot. Comparing the aggregate histograms shows that in the uniform condition participants had a flatter distribution. (B) *Shape* measure of each participant for the Gaussian and Uniform conditions, with the dashed line representing equal values for both conditions. Error bars are 95% confidence intervals (obtained via bootstrapping). Most participants had a more Gaussian sequence in the Gaussian condition (participants below the dashed line). Although most participants lie in the second quadrant (shaded), meaning that they had a Gaussian sequence in the Gaussian condition and a uniform sequence in the Uniform condition, several other participants lie in the first quadrant, meaning their sequences were Gaussian in both conditions. (C) Histogram of each participant's normalized values for each condition. For most participants, there's a clear difference between conditions, with the Gaussian values being more concentrated.

When simulating normal and uniform sequences, we found that the Type II error rate was three times smaller (0.47%) for the *Shape* measure than for our pre-registered measure (1.46%). In addition, we found the *Shape* measure more interpretable, as results from different conditions can be compared (e.g. Fig 2B).

We then ran a generalized linear mixed-effects model $Shape = \beta_0 + \beta_1 \times Distribution$ with a random intercept per participant. The average value of *Shape* was 0.18 for the Gaussian condition (SD = 0.23) and −0.05 for the Uniform condition (SD = 0.21), which constitutes decisive evidence for a difference among conditions ($t(19.00) = 4.15$, $p < .001$, $d = 1.01$, $BF_{10} = 61$). Surprisingly, several participants uttered sequences that were best fit by a Gaussian distribution in both conditions (i.e., that had a positive *Shape* value; see Fig 2B), but the vast majority of participants had a higher *Shape* value when the target was Gaussian.

## Turns at the center

In both experiments, we showed how little people changed direction compared to the *iid* expectation, as reflected by their low values of *Turning Points*. Using all items to study this

phenomenon, however, may not be fully indicative of people's sampling process. This is because participants transitioned between items in smaller jumps than *iid*, and so domain boundaries and uneven mass may not have influenced their sequences as greatly as they do in *iid* sampling. To illustrate this, in Fig 3 we plot the expected proportion of turns relative to the location of the last item in a sequence, showing that making small jumps is sufficient for low *Turning Points*: The middle row shows an *iid* sampler that was modified to never make greater transitions than one standard deviation, and as a result had a *Turning Points* value of .52 on average (as opposed to .66 for *iid*).

For these reasons, as a novel analysis, we focused our analysis on turns from the center of the uttered domain, as it is in this region—where no concerns about mass or boundaries exist—that the measure is most diagnostic. In central regions, standard local sampling models and *iid* sampling predict that the proportion of direction changes will approximately be 50%.

In Experiment 1, we restricted our analyses to items where the last utterance was between the 37.5 and 62.5 percentiles (the region of the distribution where 25% of the mass lies). This represented 26.44% and 27.63% of the data for the Gaussian and uniform condition respectively. In Experiment 2, to increase power, we limited our analysis to the three central hexes in the one-dimensional condition (37.45% of data) but this was not possible in the two-dimensional condition, where we limited our analysis to the central hex only (11.81% of data).

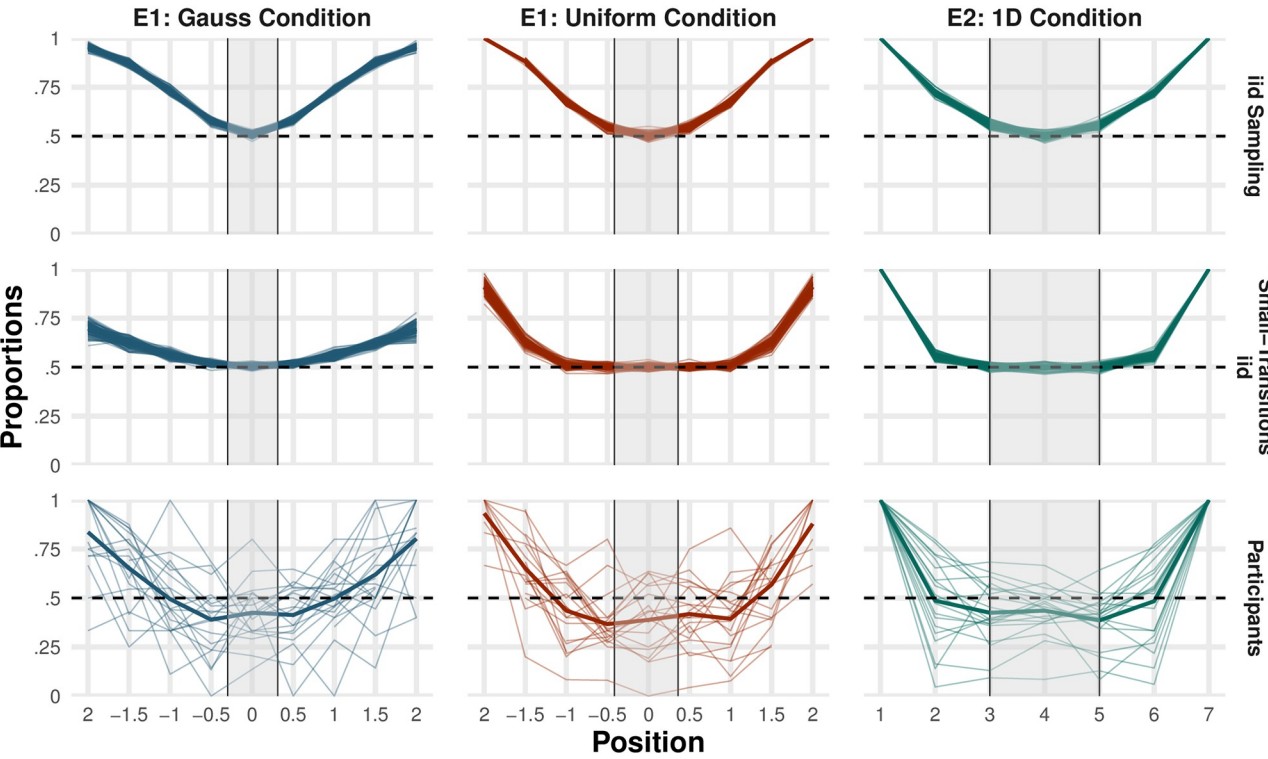

**Fig 3. Theoretical and Empirical Turn Proportions.** Proportion of turns relative to the location of the last-uttered item, for *iid* sampling (top row), a "small-transitions" *iid* sampler that discards items more than 1 SD away from the last item (middle row), and the observed values from participants (bottom row). While small *Distances* alone would lead to lower *Turning Points* values than *iid* sampling, the expected turns would be the same at the center of the distribution (i.e. around 50%). We show that people follow their current trajectory over and above what would be expected from small transitions only, going below the 50% threshold at the center of the distribution. *Note:* To be able to compare different participants in Experiment 1, we standardized their sequences and calculated a Z score of the heights uttered. Then we divided these scores into 13 possible bins, with midpoints at −3, −2.5, . . ., 0, . . ., 3, and counted the relative frequency of turns in each bin. We show bins -2 to 2 only to aid visualization. The shaded area represents the region we defined as the center in each condition. Simulated plots were generated from 100 sequences each.

In both experiments we found that people show low *Turning Points* even in this limited domain, but only in univariate domains: In Experiment 1, participants had lower *Turning Points* than *iid* (Obs. = .41, Exp. = .51, $Z = -3.37$, $p < .001$, $d = -0.34$, $BF_{10} = 4$), with no difference between the two target distributions ($Z = 2.31$, $p = .02$, $d = 0.20$, $BF_{10} = 1/3$). In Experiment 2, participant's *Turning Points* differed between conditions ($Z = 3.46$, $p = .001$, $d = 0.44$, $BF_{10} = 4$), with participants in the two-dimensional condition having *Turning Points* values similar to *iid* (Obs. = .50, Exp. = .50, $Z = 0.21$, $p = .84$, $d = 0.01$, $BF_{10} = 1/35$) but with strong evidence for smaller *Turning Points* values in the one-dimensional condition (Obs. = .41, Exp. = .54, $Z = -3.88$, $p < .001$, $d = -0.40$, $BF_{10} = 10$).

## Model comparison

### Models

We compared the performance of Cooper's schema model [36], an *iid* sampler, and six local sampling algorithms. We used Cooper's model as a stand-in for all schema models, as, to our knowledge, it is the only computational implementation of a schema account, along with Sexton and Cooper's [39] model from which it derives. As for the local sampling algorithms, we obtained them by choosing a simple algorithm (Metropolis-Hastings) and adding modifications that may approximate qualitative features of the data, as explained below (further details about models can be found in S5 Text). This expands on the sets of sampling algorithms compared in [27] and [40].

***iid* Sampling.**   This model draws independent, identically distributed random samples according to the true distribution (i.e., the ideal baseline against which people are compared).

**Schema.**   Cooper's [36] schema model, derived from Sexton and Cooper's [39] model, generates a sequence by iteratively applying one of a set of pre-learned schemas to the previous item, with some schemas being more likely than others, and with the active schema changing over time. The schema model directly applies to Experiment 1, and for Experiment 2 we made the assumption that schemas could be quickly generated for the novel representation on which participants were trained. This is a generous assumption, as it is unlikely that schemas, which are habitual in nature, could be developed for this task in such a short period of time. However, we choose to make it in order to have a computational model other than local sampling model with which to compare participants' data with.

**Local Sampling.**   We include six Markov Chain Monte Carlo (MCMC) algorithms in our comparison; a family of algorithms that are widely used in statistics [41], and which have previously been compared to human data before [27, 30, 31, 33, 42]. They operate by creating a chain of states that are only dependent on the previous one. In each iteration, an update to the current state is proposed, with more likely states being favored (thus the chain approximating a distribution).

The details of how state updates are proposed and accepted is what differentiates MCMC algorithms. We use Metropolis-Hastings [43] as the base algorithm: in each iteration, it proposes a new state by adding random noise, then it evaluates whether to transition to that proposed state based on the likelihood ratio of the proposed and current locations, with a preference for more likely locations.

We create the other five local sampling algorithms by adding qualitative features to Metropolis-Hastings in a semi-factorial way (see Table 1):

- *Multiple chains*: This feature involves running multiple chains that swap places stochastically, with some chains transitioning to lower-likelihood states more frequently. This makes

**Table 1. Qualitative features of the local sampling algorithms compared.**

| | Samplers | | | | | |
|---|---|---|---|---|---|---|
| Qualitative Modifications (# Parameters) | MH | MC³ | HMC | REC* | MCHMC* | MCREC* |
| Multiple chains (3) | ✗ | ✓ | ✗ | ✗ | ✓ | ✓ |
| Gradient-based proposals (1) | ✗ | ✗ | ✓ | ✓ | ✓ | ✓ |
| Recycled momentum (1) | ✗ | ✗ | ✗ | ✓ | ✗ | ✓ |

Sampler abbreviations are MH: Metropolis-Hastings, MC³: Metropolis-coupled Markov Chain Monte Carlo, HMC: Hamiltonian Monte Carlo, REC: Recycled-momentum Hamiltonian Monte Carlo. MCHMC and MCREC are the Metropolis-coupled versions of HMC and REC. Starred algorithms are those that have not been compared to human data in past work. Which parameters govern a sampler's behavior is determined by these qualitative features (see S5 Text for details).

exploration more efficient in multimodal domains (as the sampler is more likely to traverse low-likelihood valleys).

- *Gradient-based proposals*: This feature involves having samplers propose new states in a way that utilizes the gradient of the posterior distribution, by simulating a physical system using Hamiltonian dynamics, where the current state is the position and the momentum is drawn randomly in every iteration [44]. This makes the sampler more efficient in multi-dimensional domains.

- *Recycled momentum*: This feature changes how the momentum is chosen in each iteration of gradient-based samplers, obtaining it by partially 'recycling' the previous momentum rather than drawing it randomly. This feature can only be added if proposals are gradient-based (hence the semi-factorial design), and can be used to make exploration more directed (avoiding the back and forth of random walks).

We generated $10^5$ sequences for each model, each time drawing parameters from a uniform prior. For Experiment 1, where the true distribution is unknown, we estimated the best-fitting Beta or Normal distribution for each participant; *iid* and local sampling models sampled from a distribution defined by the average parameters of participants' best fitting distributions. The schema model uses the range of possible responses to sample, which we obtained by calculating the average minimum and maximum value of participants' sequences.

To compare people's sequences to the performance of these generative models, we used Approximate Bayesian Computation (ABC, [45]), a simulation-based technique that can be used to perform model comparison in cases where no likelihood function is available (see Methods). We used the above summary measures to compare these models to people's performance, restricting *Turning Points* to the center of the distribution as defined above.

Because Cooper's model assumes the distribution to be uniform, the author also uses measures of entropy in his analyses. To ensure that our comparison is fair, we also analyze the sequences from uniform distributions using the entropy measures used in Cooper [36], but for simplicity, and because results are qualitatively the same, we report those analyses in S7 Text. Because the schema model cannot sample in two-dimensional domains, and because this condition proved undiagnostic between local sampling features, we also relegate the discussion of the different models in the two-dimensional condition of Experiment 2 to S6 Text.

## Results

**Experiment 1, Uniform condition.** The average best fitting distribution was *Beta*(1.27, 1.43), and the average range of possible values was from 122 to 219 cm (97 possible values).

Model recovery results were good, with the *iid* and schema models being correctly classified 99% and 95% of the time respectively, and with local sampling models being correctly classified 86% of the time. REC was misclassified as HMC and MCREC as MCHMC 21% and 27% of the time respectively: This is to be expected somewhat, as the behavior of models with recycled momentum becomes more similar to models without it the lower the amount of recycling is.

Local sampling algorithms approximated participants' *Shape* best, while the schema model generated consistently more uniformly-distributed sequences. All models produced too small *Adjacencies* and too large *Distances* overall, and only the gradient-based samplers with multiple chains (MCHMC and MCREC) and the schema model could approximate participants' *Distances* in some simulations. All samplers but the schema model produced too large values for *Repetitions*, and only the samplers with recycled momentum (REC and MCREC) and the schema model could consistently match people's low *Turning Points* (see Fig 4 for distributions of summary values).

Quantitatively, local sampling algorithms performed better than the *iid* ($BF_{10} = 2.0 \times 10^{105}$) and schema ($BF_{10} = 5.1 \times 10^{30}$) models in this condition, and predicted 14 out of 20 participants best (the schema model predicted 6 participants best). Regarding local sampling qualitative features, we found support for models running multiple chains ($BF_{10} = 2.2 \times 10^{31}$), having gradient-based proposals ($BF_{10} = 1.4 \times 10^{43}$), and recycling their momentum ($BF_{10} = 8.0 \times 10^8$; see Fig 5, first column).

**Experiment 1, Gaussian condition.**   Models had a Gaussian distribution with a mean of 176.4cm and a standard deviation of 12cm as the target (irrespective of whether the participant sampled male or female heights), which were the average parameters of participants' individual best fitting distributions. The estimated range of responses (used by the schema model) was smaller here, 83cm (from 131cm to 213cm). Model recovery results were good: the *iid* and schema models were correctly categorized 99.9% of the time, and local samplers were correctly categorized 78% of the time, with REC being misclassified as HMC and MCREC as MCHMC 19% and 21% of the time respectively.

All models except the schema model (see Fig 4) could replicate the fact that participants reproduced the target distribution in the Gaussian condition (similar *Shape* values). Here, local sampling algorithms matched participants' *Repetitions*, but this was more due to an increase in people's frequencies than to a change in sampler behavior (on average, people's *Repetitions* were .07 in the Gaussian condition and .02 in the Uniform condition; $Z = 10.05$, $p < .001$, $d = 0.98$, $BF_{10} = 6.8 \times 10^8$). Again, only samplers with recycled momentum and the schema model matched people's low *Turning Points*.

Once more, local sampling algorithms replicated participants' data better than the *iid* ($BF_{10} = 2.1 \times 10^{105}$) and schema ($BF_{10} = 3.7 \times 10^{120}$) models, and predicted 15 participants best (with *iid* and schema predicting 2 and 3 participants best respectively). As for qualitative local sampling features, we found support for running multiple chains ($1.3 \times 10^{21}$), using gradient-based proposals ($3.9 \times 10^{25}$) and recycling momentum ($BF_{10} = 4.6 \times 10^5$; see Fig 5, second column).

**Experiment 2, One-dimensional condition.**   The average best fitting distribution was *Beta*(1, 1) (i.e., a uniform distribution), and so this was the target distribution for both the *iid* sampler and the local sampling algorithms. Model recovery results were poor for the *iid* model and the local sampling models: while the prior error rate for schema models was 27%, the *iid* model and local sampling models had error rates of 64% and 82%, with *iid* being misclassified as a local sampling model 63% of the time, and local sampling models being misclassified as another local sampling model 52% of the time.

All models were able to consistently replicate participants' *Shape* values, but their values for *Adjacencies* were too low and their values for *Distances* too large. Regarding *Turning*

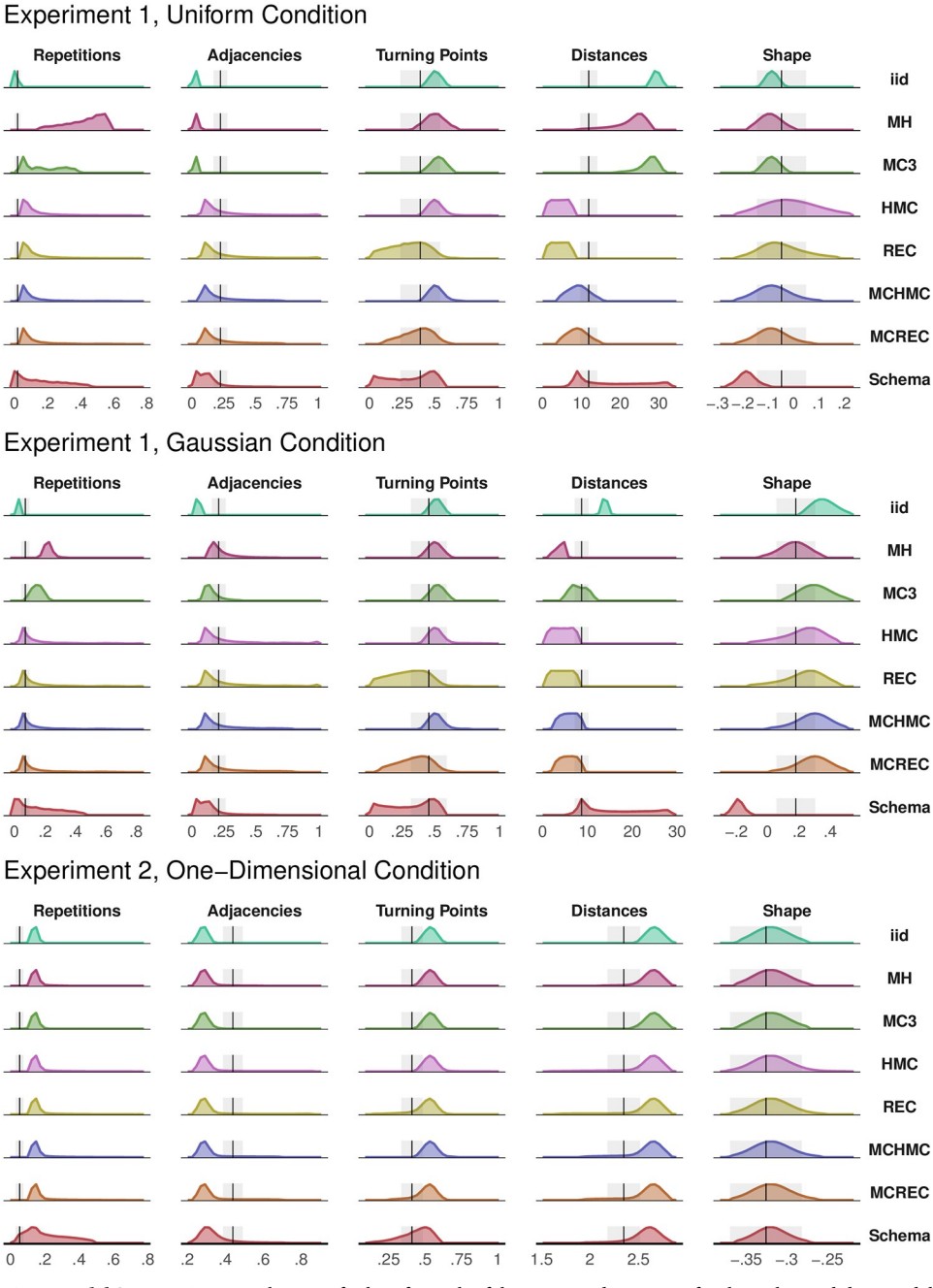

**Fig 4. Model Summaries.** Distribution of values for each of the computed measures for the eight candidate models, in three tasks, with parameter values drawn from the prior. Further details are provided in the main text. Notice that in the Gaussian condition of Experiment 1 the *iid* sampler had fewer repetitions than participants, yet in the main text we reported that people had fewer repetitions than expected. This is because there we compared their performance to reshuffled sequences, not to *iid* sampling from the distribution, and people have fewer unique items than *iid* sampling.

*Points*, only the local sampling algorithms that recycled their momentum (REC and MCREC) and the schema model replicated the low number of *Turning Points* at the central hex that people produced. Finally, only the schema model had as low *Repetitions* as participants (see Fig 4).

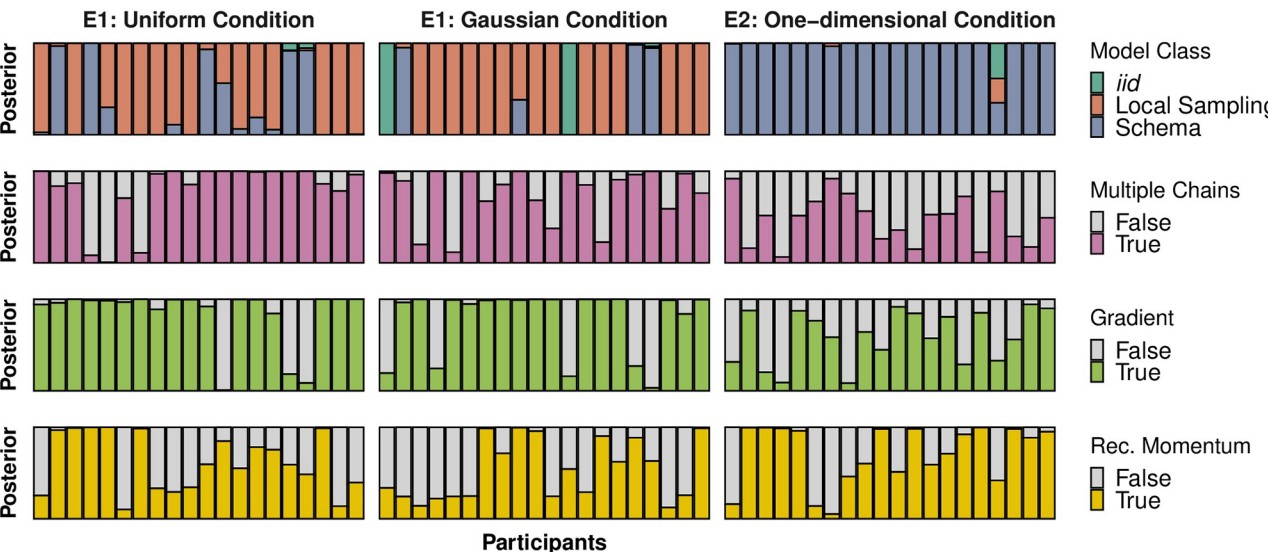

**Fig 5. Posteriors per participant.** Posteriors per participant in three tasks, with each column representing one participant. While in the one-dimensional condition the schema model performed best, local sampling algorithms replicated participants' data better in the Uniform and Gaussian conditions. Within local sampling features, we found decisive evidence for samplers running multiple chains, using gradient-based proposals, and recycling their momentum. *Note*: Conditions in Experiment 1 varied within participants, and so the *n*th bar in the Uniform condition is the same participant as the *n*th bar in the Gaussian condition.

For these reasons, the schema model was closest to people's performance in this condition, predicting 19 out of 20 participants best (with *iid* predicting one participant best), and with a $BF_{10} = 9.2 \times 10^{89}$ over the local sampling class of models and $BF_{10} = 3.6 \times 10^{136}$ over the *iid* sampler. However, this is a generous interpretation of the schema model, assuming that people are able to very quickly learn and apply schemas to novel representations.

Despite the schema model outperforming local sampling models in this condition, we still compared the qualitative features of local sampling algorithms: we found evidence against models running multiple chains ($BF_{10} = 4.0 \times 10^{-5}$), evidence for gradient-based proposals ($BF_{10} = 2.7 \times 10^{3}$), and evidence for recycled momentum ($BF_{10} = 4.8 \times 10^{16}$; see Fig 5, third column), although the model recovery results for this condition reveal that local sampling models were not particularly distinctive among themselves.

**Combined Posterior Probabilities.**   Finally, we combined the obtained posterior probabilities for the three conditions by multiplying them together, in order to carry out joint comparisons. When aggregating over conditions, we found decisive evidence for local sampling algorithms over the schema model ($BF_{10} = 2.9 \times 10^{58}$). We also found decisive evidence for local samplers running multiple chains ($BF_{10} = 1.3 \times 10^{48}$), using gradient-based proposals ($BF_{10} = 1.4 \times 10^{85}$), and recycling their momentum ($BF_{10} = 7.2 \times 10^{26}$).

Comparing the individual models across all three conditions, we found that the best fitting model was MCREC, with a $BF_{10} = 7.2 \times 10^{26}$ over the next best model, MCHMC, and a $BF_{10} = 1.8 \times 10^{59}$ over the schema model (see Table 2).

## Discussion

In the current study, we expanded the random generation paradigm to ask participants to generate sequences from non-uniformly distributed domains and from recently-learned displays. We found that participants displayed the same systematic deviations from *iid* sampling as

**Table 2. Model Bayes Factors.**

| Model | E1: Unif Condition | E1: Gauss Condition | E2: 1D Condition | Combined Posteriors |
|---|---|---|---|---|
| MCREC | $1.2 \times 10^{106}$ | $1.2 \times 10^{106}$ | $9.4 \times 10^{42}$ | $1.4 \times 10^{255}$ |
| MCHMC | $1.5 \times 10^{97}$ | $2.7 \times 10^{100}$ | $4.9 \times 10^{30}$ | $2.0 \times 10^{228}$ |
| REC | $5.4 \times 10^{74}$ | $8.6 \times 10^{84}$ | $2.4 \times 10^{47}$ | $1.1 \times 10^{207}$ |
| Schema | $4.0 \times 10^{74}$ | $5.6 \times 10^{-16}$ | $3.6 \times 10^{136}$ | $8.0 \times 10^{195}$ |
| HMC | $1.2 \times 10^{59}$ | $9.2 \times 10^{83}$ | $3.9 \times 10^{25}$ | $4.2 \times 10^{168}$ |
| MC$^3$ | $1.1 \times 10^{54}$ | $6.9 \times 10^{74}$ | $1.9 \times 10^{14}$ | $1.4 \times 10^{143}$ |
| MH | $7.5 \times 10^{-15}$ | $6.1 \times 10^{21}$ | $1.8 \times 10^{27}$ | $8.2 \times 10^{34}$ |
| *iid* | 1 | 1 | 1 | 1 |

Bayes Factors for the eight candidate models, in each separate task and considering the joint posterior probabilities. Models have been arranged in order of Bayes Factor over *iid* when combining the three tasks, from largest to smallest. Metropolis-coupled Recycled-momentum Hamiltonian Monte Carlo (MCREC) is the best-fitting model, but other local sampling algorithms are also competitive.

found in previous random generation experiments, pointing to a common mechanism underlying their performance across tasks. We also showed that participants could flexibly change the distribution from which the generated samples, being able to generate the same items—heights, in our case—in a uniform or Gaussian fashion depending on the given instructions. Finally, we identified a key qualitative feature of people's random generation, people's tendency to maintain their trajectory for many samples, and showed that this pattern does not arise only due to the fact that people make small transitions. These findings directly contradict schema accounts of human random generation [16, 36], which predict that people generate random items by striving for serial independence only, and that the systematic deviations from random sampling that people display arise from the presence of habitual responses (schemas) and the attempt to suppress them. The schema account could be relaxed to allow for quick learning of schemas (as we assumed when modeling the one-dimensional condition of Experiment 2), or more sophisticated and task-specific schemas could be postulated for the domains examined here. However, these hypothetical schema accounts would still not reproduce target distributions, and this account would need substantial modifications in order to do so: for example, additional monitoring processes on the response histogram, not its unpredictability, would need to be included, as well as a policy on how to trade off deviations from unpredictability and from the optimal histogram.

Although no computational model was available for the network modulation account of random number generation [23], the results here presented can qualitatively be compared to what this account would predict. The network modulation account postulates that people generate random items by creating an associative network with each possible response as a node, and with links representing their strength of association. While this network alone produces habitual, stereotyped responses, a controller inhibits the strongest links to allow for more unpredictable behavior.

If it were modified to have bidirectional links between items, with the strength of each direction being proportional to the density ratio between the receiving node and the origin, this account would be able to reproduce the fact that people can generate random items in a Gaussian fashion. Reproducing the finding that people make fewer turns than expected would require a very specific kind of interaction between the associative network and the controller: because the associative network cannot produce fewer turns (as transitions only depend on the current state and not the previous state), the controller would have to modify the network's links frequently and in a very distinct way in order to achieve this result. In addition, this

account would also not be able to explain the fact that the same deviations from *iid* sampling appear when the items have recently been learned, as no habitual responses need to be inhibited in this case.

Instead, data are consistent with the alternative we proposed: that people are not using a surrogate process to sample items, but instead sampling from the domain directly using their general cognitive ability to produce samples to perform inferences. We also quantitatively modeled people's behavior to the performance of a schema model [36], an *iid* sampler, and six MCMC algorithms that varied in three qualitative features. In one dataset—the one-dimensional condition of Experiment 2—the schema model performed best. This was due to it being able to reproduce the low level of repetitions participants displayed in such a small range of possible items, as such a range is closest to the tasks it was designed for. The fact that we assumed that schemas could be applied to newly-learned items, however, should be noted when interpreting these results. In the other three datasets, as well as when combining posteriors across datasets, we found that local sampling algorithms were best at replicating human performance, thus linking people's behavior in random generation tasks to their performance in other domains. We identified several features, such as recycling momentum, that had not been previously compared to human data, showing that these features allowed local sampling algorithms to better fit the data than the sampling algorithms that have performed best in past comparisons (i.e., $MC^3$ [27, 40]).

Recent research using process-tracing techniques suggests that how samples come to mind and what the task at hand is are largely independent. For example, Mills and Phillips [46] ask their participants to generate a list of animals as they come to mind, and find that doing so with no other purpose, or in order to answer a specific question, does not change the types of items participants produce. Similarly, Hardisty, Johnson and Weber [47] find that reporting ideas that come to mind while making a decision makes no qualitative difference on the resulting choice, compared to a condition without thought listing.

If how samples are accrued is task-independent, then having identified these local sampling features in the random generation task also has implications for how people engage in the other tasks where local sampling algorithms have been applied: These sampling approaches have been used to explain how people come up with ideas in a semantic fluency task [33], how they estimate temporal duration [27], how they perceive visual stimuli with multiple interpretations [8] and why they present multiple biases in how they reason with probabilities [28, 30, 31].

Although more and more research is being done comparing human performance to local sampling algorithms, the set of available algorithms in the computer science literature is incredibly vast, and efforts to identify the features of the human mental algorithm are in their infancy. An exciting conclusion of the current work is that random generation tasks can distinguish fine-grained differences between algorithms, an endeavor that can be expanded on in future work.

### Limitations and Future Directions

Despite their success, local samplers displayed too high *Repetitions* in most conditions. Future research may investigate additional qualitative features that can improve on this fact: for example, 'unadjusted' algorithms [48], which do not evaluate the relative goodness of the proposed and last state before transitioning, would repeat less by not rejecting proposals. Alternatively, a post-sampling mechanism that explicitly eliminates some of the repetitions could be implemented, akin to participants choosing not to utter their sampled item if it is identical to the last; or sequences could be thinned to only report the *n*th sample, following research suggesting that people use few, but more than one, samples when making probability judgments [6].

An exciting feature of the current data is that people were able to replicate the target distribution in Experiment 1, despite never being told participants what the true distribution of heights in the United Kingdom is. Another avenue of future research, therefore, is to use random generation as a belief elicitation method. In many domains, knowing what experts believe is essential to build models of possible future outcomes [49], and in related work, we have shown that random generation can be used to elicit beliefs as a complement to other more established techniques [50]. This was true for additional distributions to the ones shown here: distributions with high skewness (gross earnings from films) or where some values are extremely unlikely (American football scores).

Another avenue of future research will be to apply local sampling models to previous random generation results: most notably, random generation research on neurodivergent populations might benefit from a sampling interpretation. A vast literature has shown that different neurodiverse populations show differences with neurotypical controls for some measures of randomness but not others. For example, patients with schizophrenia and Parkinson's disease will display even higher rates of adjacent items than neurotypical controls but the same bias against repeating items [51, 52], while patients with multiple sclerosis will make even fewer turns than healthy controls [14]. Conversely, patients with unilateral frontal lobe lesions may show a lesser bias against repeating items than neurotypical adults [53]. These differences can be framed as differences in the qualitative features of the sampling model or as differences in the parameter values used. For example, a local sampling model without multiple chains will have more adjacent items, and increasing the degree of recycled momentum will lead to fewer turns.

The models here presented might also need to be expanded to account for the many previous findings on how neurotypical adults perform the random generation task. For example, many studies have manipulated how participants input their responses: rather than say items out loud participants may press keys on a keyboard [54], select items with a mouse [18], or fill squares in a grid [55], with differences in how random resulting sequences are. Analogously, changing how the task of being random is described to participants may influence their sequences (for example, asking them to simulate a coin toss mentally will lead to more random sequences than explicitly asking for a random choice between heads and tails [56]). Participants are also more random when participating in a zero-sum game like matching pennies or rock paper scissors [20, 57]. While it is possible that changes in the input mechanism may influence the mental representation participants have of the task (in the same way that our spatial training of syllables did), and that different instructions or a competitive setting may result in differences in effort while doing the task, future work will be needed to incorporate these findings into local sampling models.

It is possible, however, that some other findings in the random generation literature not here explored would be already replicable by the local sampling models we use in this study: previous research has shown that people are less random and say more adjacent items when production rates are high [16, 21, 58], which could be achieved by a local sampling model that makes smaller proposals between utterances. Similarly, if people have a second task to perform while they're generating a random sequence, they display even fewer repetitions and fewer turning points [18], which a model with an increased degree of recycled momentum would reproduce.

## Conclusion

We devised novel tasks to study volitional random generation and found that people can generate random items from a wider range of domains than previously studied, while still

displaying the characteristic deviations from *iid* sampling observed in previous tasks. We showed that local sampling algorithms replicate people's data more successfully, linking people's randomness in the random generation paradigm to behavioral noise in other tasks. We were also able to identify several qualitative features that mirror people's sequences, showing that random generation can be a useful paradigm to reverse-engineer people's sampling algorithm.

## Methods

### Ethics statement

For both experiments, ethical approval was given by the Humanities and Social Sciences Research Ethics Committee (HSSREC) at the University of Warwick. Written informed consent was obtained from all participants.

### Experiment 1

In this experiment (preregistration at https://osf.io/ux5tp), we asked participants to produce a random sequence of people's heights, either from a distribution in which all possible heights are equally distributed, or from the actual distribution of heights in the United Kingdom (which is roughly Gaussian for both adult men and adult women [37]). All participants generated heights from both distributions and the order of the distributions was counterbalanced.

**Participants.** Participants were recruited from the University of Warwick participant pool. Being fluent in English was the only inclusion criterion. To ensure that participants accessed the task with an appropriate microphone and stable internet connection, a pre-screening task was run in which candidates recorded themselves reading a short text. Participants were paid £0.50 for completing the pre-screen task, irrespective of the outcome. The first two authors independently rated whether the recording was audible, and the participants were invited to perform the main task if at least one rater had deemed their recording to be valid. Raters reached high agreement (87.2%, Cohen's $\kappa$ = .63). 85% of the participants who participated in the pre-screen were invited to the main experiment. We collected data from 21 participants (Mean Age = 23.8, SD = 4.71; 14 male, 7 female). Following pre-registered criteria we calculated a measure of how predictable sequences were (sequence determinism [59], see S9 Text]), and data from one participant (5%) was excluded from analysis (91% of their items were one inch taller than the previous). Participants received payment of £2.5 plus a bonus of up to £1.35 depending on performance, which we measured by how well they kept to the given pace (mean total payment = £3.71). The experiment lasted approximately 20 minutes.

**Design and Procedure.** The experiment was conducted via video call. First, we introduced participants to the task, and then they had to produce random heights from that distribution for five minutes. In one condition, heights were introduced as distributed according to a uniform distribution, whereas in the other condition, heights followed the true distribution for adults in the UK for the target gender. Following [37], we consider the true distribution of UK heights to be Gaussian for each gender, with a mean of 176.4cm and standard deviation of 7.02 for men and a mean of 163.6cm and a standard deviation of 6.03 for women (we did not disclose this information to participants). We will refer to these conditions as Uniform and Gaussian, respectively.

Participants produced heights at random in two blocks, one for each condition, in counterbalanced order. The target gender was fixed for each participant throughout the experiment. Participants could express heights in feet and inches or meters and centimeters, and we analyzed all randomness measures with the units they had used (For simplicity, we use centimeters only throughout the current text).

At the beginning of the experiment, participants were asked what they believed the height of the shortest and tallest adult was in the UK (for the gender in their condition). Some participants spontaneously asked whether they should consider people with restricted growth, which they were told not to. Participants estimated the minimum and maximum values reasonably well, below and above the 1st and 99th percentiles of the true distribution: their median minimum was 136 cm (SD = 17) for men and 125 cm for women (SD = 23.6), and their median maximum was 217 cm for men (SD = 20.3) and 200 cm for women (SD = 35.9).

After these preliminary questions, participants were told a cover story matching the experimental condition for that block. In the Gaussian condition, participants were asked to imagine that a photographer wanted to take a picture representing the heights of adults (of the gender they had been allocated), taking a picture of 10,000 people so that each possible height appeared as often as it does in the population (as in the 'living histograms' of [60]). They were told to imagine that each person who had been in the picture wrote their height on a piece of paper, and that that paper was put in a bag. In the Uniform condition, they were told to imagine that each possible height within the height boundaries they had previously specified had been written on a piece of paper and that all papers were put in a bag.

After the learning stage, participants were asked to produce random heights, as if they were drawing a random paper from the bag that had been described, saying the height out loud, putting the paper back in the bag, and reshuffling the papers. They repeated this process for five minutes. While saying items out loud, they were asked to look at the screen, where a dot flashed at 30 times per minute, and were instructed to say a height every time the dot appeared. The pace of production was chosen to be slower than in Experiment 1 because pilot testing revealed that participants required more time to utter the multi-syllable heights.

After five minutes, participants were allowed a short rest, and then the Learning and Production stages were repeated for the other condition. Participants produced both tasks at a similar pace—the median temporal gap between successive items was 2.09s (SD = .27) and 2.02s (SD = .24) for the first and second sequence participants produced, a difference that was significant but ambiguous ($F(1, 19) = 5.55$, $p = .03$, $d = −0.53$, $BF_{10} = 1$).

## Experiment 2

Preregistered (https://osf.io/q3yrj) analyses for this experiment were reported in [61]. Analyses here followed the analysis plan for Experiment 1.

In this experiment, participants first learned a display of syllables, arranged in either a one-dimensional row or a two-dimensional configuration (see Fig 1), by moving virtually through the display, revealing the syllable at their current location. Then, once they could reproduce these spatial arrangements from memory, they were asked to utter a random sequence of syllables from that set.

**Participants.**   We recruited 42 participants (Mean Age = 24.95, SD = 9.67; 14 male, 27 female, 1 non-binary) from the University of Warwick participant pool. The only inclusion criterion was that participants had English as their first language. This was a stricter language requirement than in Experiment 1, as we wanted to ensure that the syllables were meaningless to participants. Two participants (5%) did not learn the syllables in the allocated time, and were excluded from analysis following our preregistered criteria. Participants received a payment of £3.5 plus a bonus of up to £1.8, which depended on their performance in learning the syllables (the average total payment was £4.64). The experiment lasted approximately 30 minutes.

**Materials.**   We chose seven syllables of two letters each, all ending in *a* to ensure consistent ease when uttering consecutive items. To select them, we considered both the frequencies of

the syllables and syllable pairs in the Brown corpus [62], aiming for a homogeneous set. The resulting selection was: *ca*, *ha*, *la*, *ma*, *na*, *pa*, and *ta*.

**Design and Procedure.**   The experiment was conducted via video call. Participants first learned the display they had been allocated (Learning Stage), then they uttered syllables at random (Production Stage) for five minutes. The two blocks, learning the display and producing a random sequence, were repeated after a short break (with the same display in both blocks). The key experimental manipulation, which varied between participants, was whether the display of syllables they learned was the one- or two-dimensional arrangement. How the syllables were arranged within the one- or two-dimensional display followed one of five possible configurations and was chosen randomly for each participant.

In the learning stage, participants were presented with a display consisting of seven hexagons, arranged in either a single row or a two-dimensional grid, depending on the experimental condition. The hexagons were oriented so that the vertex was on top, and the two-dimensional grid consisted of three rows of two, three, and two hexagons, respectively. Each hexagon contained a hidden syllable, and participants' task was to view and learn which syllable each hexagon displayed. To do so, they selected a hexagon whose syllable they wanted to reveal, which made the previous syllable disappear, and the syllable in the chosen hexagon appeared. They could freely choose any hexagon as their starting one, but subsequent choices were constrained to adjacent hexagons only, which made the learning process akin to 'spatially exploring' the display. To promote active learning, we included a delay of one second between the disappearance of a syllable and the appearance of the next, and instructed participants to announce which syllable they expected in the hexagon they had selected before it appeared.

As soon as participants felt confident that they had learned the display, their knowledge was tested by asking them to name the syllables displayed on the seven hexagons in random order. If participants answered all seven queries correctly in two consecutive tests, they proceeded to the production stage, or else they returned to learn the display. Participants were excluded, and the experiment was terminated if they failed the test four times, or if they exceeded the maximum learning time of 10 minutes. Participants spent an average of 5.7 minutes (SD = 2.4) learning the syllables, and no participant spent more than ten minutes. Two participants failed the test four times and were excluded (on average, participants failed 0.65 times, SD = .86. The two excluded participants' average learning time was 9m and 54s).

In the production stage, participants uttered syllables from the set they had learned at random for five minutes. To instruct participants to produce random sequences, we asked them to imagine that they were drawing the syllables out of a hat each time, and putting the syllable back before shuffling and drawing the next, following standard practice in previous random generation experiments [16, 17]. During this stage, participants did not see the display they had learned, but instead saw a dot flashing on screen, appearing at a pace of 80 times per minute (once every 750ms). Participants produced a slightly lower pace than targeted (M = 71.21 syllables per minute, SD = 14.32).

After completing this stage, participants were allowed a short break. Then, both learning and production stages were repeated, using the same display of syllables in the same arrangement: participants had the opportunity to revise the display, and after testing they uttered syllables for another five minutes. The average time spent revising and testing was 104s (SD = 47s), and the average number of failed attempts was 0.18 (SD = .38). Both were much lower than in the initial learning stage ($t(39) = -11.61$, $p < .001$, $BF_{10} = 1.2 \times 10^7$ and $t(39) = -3.48$, $p = .001$, $BF_{10} = 9$, respectively), which suggests that participants had no difficulty remembering the display after their first random generation block. Participants were slower at producing items in the first block they produced, with the median temporal gap between the items they uttered being larger for the first block participants produced (median = 843ms, SD = 146) than for the

second (median = 799ms, SD = 142). The evidence for this difference was ambiguous, with a significant p-value but anecdotal Bayes factor ($F(1, 39) = 11.97$, $p = .001$, $d = -.33$, $BF_{10} = 2.46$). All participants named each syllable at least once in each sequence.

## Model comparison

**Method.**    To compare people's sequences to the performance of the several generative models, we used Approximate Bayesian Computation (ABC [45]), a simulation-based technique that can be used to perform model comparison in cases where no likelihood function is available. In our case (many approaches to ABC are available), we first generated vast volumes of artificial data from the candidate models, sampling random values for their parameters from their prior distributions; then obtained summary measures for the observed and synthetic data. Finally, we used a machine learning tool for data classification, random forests, to compute the posterior belief on each of the candidate models [63]. In short, a set of classification decision trees is trained on the simulated data and learns to categorize it into the different candidate models it originated from. Then, it classifies the observed data into which candidate model is more likely to have produced it. To obtain the probability that the classification model makes an error, a separate regression forest is trained on the same training data and the classification error during training, and is applied to the observed data. This regression forest only computes the probability that the classification label provided was incorrect, and so to obtain a full posterior distribution over all candidate models, we ran forests recursively for each uttered sequence, each time removing from the candidate models the model that had been considered best in the previous iteration, until a posterior for each model was obtained. Unlike more traditional ABC approaches, the random-forest approach requires fewer simulations for each candidate model, and is more robust to the choice of summary statistics (for completeness, however, we carry out a more typical ABC analysis in S8 Text, with similar results).

Here, we fit canonical distributions to participants' data in each experiment, which would be the target distributions the models would sample from. We then simulated $10^5$ sequences of 400 items each for each candidate model and each condition, choosing the prior over the parameters to be as uninformative as possible (we describe model parameters and their associated priors in S5 Text). Because the *iid* sampler and the local sampling algorithms sample in continuous space, but participants produced whole numbers, we rounded the values the samplers produced. To evaluate the resulting data, we used the summary statistics described in the main text, choosing to focus on *Turning Points* from the central region of the distribution. Because the schema model can only sample from univariate distributions, we do not consider the two-dimensional condition of Experiment 2 in the main text, but we did fit local sampling models there too, and we describe those results in the S6 Text (this subset of the data, however, yielded uninformative results, with all local sampling algorithms and the *iid* sampler performing equally well).

In order to compare the three higher-level approaches to random generation (*iid* sampling, local sampling, and schema accounts), we carried out Bayesian Model Averaging [64] to compute Bayes factors (i.e., the ratio of the average posteriors of each candidate class). We also used these to compare qualitative features of the candidate local sampling models, comparing models that were 'matched': we only included models to our comparisons that had an identical equivalent in the other side of the comparison but for the factor of interest (e.g., to compute Bayes factors of inclusion for gradient-based proposals, the average of the posteriors for HMC and MCHMC was compared to that of MH and MC$^3$. Because no model exists that has recycled momentum but not gradient-based proposals, REC was not added to either side of the comparison; see Table 1).

## Supporting information

**S1 Text. Additional Exploratory Plots.**
(PDF)

**S2 Text. Condition Effects.**
(PDF)

**S3 Text. Order Effects.**
(PDF)

**S4 Text. Other pre-registered analyses.**
(PDF)

**S5 Text. Model Details.**
(PDF)

**S6 Text. Model comparison in the two-dimensional condition of Experiment 2.**
(PDF)

**S7 Text. Entropy Measures.**
(PDF)

**S8 Text. Standard Approximate Bayesian Computation.**
(PDF)

**S9 Text. Exclusion Criteria.**
(PDF)

## Author Contributions

**Conceptualization:** Lucas Castillo, Pablo León-Villagrá, Nick Chater, Adam Sanborn.

**Data curation:** Lucas Castillo.

**Formal analysis:** Lucas Castillo.

**Funding acquisition:** Adam Sanborn.

**Investigation:** Lucas Castillo.

**Methodology:** Lucas Castillo, Pablo León-Villagrá, Nick Chater, Adam Sanborn.

**Project administration:** Lucas Castillo, Adam Sanborn.

**Software:** Lucas Castillo.

**Supervision:** Adam Sanborn.

**Visualization:** Lucas Castillo, Pablo León-Villagrá.

**Writing – original draft:** Lucas Castillo, Pablo León-Villagrá, Adam Sanborn.

**Writing – review & editing:** Lucas Castillo, Pablo León-Villagrá, Nick Chater, Adam Sanborn.

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
