## [Decision Letter · Decision Letter 0]

18 Aug 2023

Dear Mr Castillo,

Thank you very much for submitting your manuscript "Explaining the Flaws in Human Random Generation as Local Sampling with Momentum" for consideration at PLOS Computational Biology.

As with all papers reviewed by the journal, your manuscript was reviewed by members of the editorial board and by several independent reviewers. In light of the reviews (below this email), we would like to invite the resubmission of a significantly-revised version that takes into account the reviewers' comments.

Please note that the reviewers especially highlighted issues with framing the study within the previous literature, as well as problems with testing the analysis code.

We cannot make any decision about publication until we have seen the revised manuscript and your response to the reviewers' comments. Your revised manuscript is also likely to be sent to reviewers for further evaluation.

Sincerely,

Ulrik R. Beierholm

Academic Editor

PLOS Computational Biology

Daniele Marinazzo

Section Editor

PLOS Computational Biology

Reviewer's Responses to Questions

**Comments to the Authors:**

Reviewer #1: Uploaded as attachment

Reviewer #2: Summary

======

This manuscript is part of a larger effort by the authors to account for systematic deviations from what might be expected from a more perfect cognitive system by postulating that the brain samples locally. Using that approach, they strive to reconcile what might be important, though divergent, models of cognitive function (e.g., Chater et al., 2020, for example). In this manuscript they take aim at random-pattern generation, striving to explain both why humans act more randomly than they should when they need not be random and why they are not random enough when specifically asked to be random. In particular, the authors claim that their model is better than the schema theory that, they say, currently prevails as an explanation for human random-pattern generation. The authors carry out two experiments, one of which was previously reported, to test this hypothesis. And they show that their results favor their model of local sampling with momentum. The application of their model to random-number generation is interesting and potentially fruitful. And the authors generally provide good evidence for it (though see below). But the manuscript suffers from several problems that are articulated below.

Major comments

The main drive of this study is pitting the authors’ theory of random number generation against the schema theory, suggested by (Baddeley, 1996) and formalized by Cooper (2016). However, this theory is not as central in the field as the authors make it seem. In fact, many of the papers that they cite when discussing results in the field of random-number generation do not even mention schema theory. What is more, other models for random-number generation have been proposed, like Jahanshahi’s more brain-based Network-Modulation Model, which the authors do not mention. So, aiming to provide evidence specifically against the schema theory to promote their own might be somewhat of a strawman argument.

What is more, it is unclear how strong that evidence is, or, in other words, whether the final conclusions about the schema vs MC samplers are warranted. This is because the schema model outperformed the other MC samplers on Experiment 2’s one-dimensional data. Maybe adding a discussion on why MCREC did better on the aggregate data but not the individual datasets could help here.

In Experiment 1, the authors note that they chose to test the participants on a Gaussian (or normal) distribution rather than a uniform one, because that is a key differentiating factor between schema theory and theirs (lines 81-90 and 175-176). However, they could do more to explain why the schema theory does not allow for non-uniform distributions. While their claim may well be true, some simple extensions of the schema theory could make it better fit non-uniform distributions too. In sum, they should do more to explain why schema theory does not fit with nonuniform distribution to make it easier for the reader to understand. What is more, they wanted to test random-series generation in non-uniform settings. However, the Gaussian distribution is rather similar to the uniform one; ~⅔ of the weight lies within 1 STD from the mean, ~95% lies within 2 STDs from the mean, and almost all of it lies within 3 STDs from the mean. The authors use measures to specifically compare Gaussian to uniform distributions (Fig. 2). But to better argue for the ability of humans to generalize to non-uniform distributions, and to potentially make a better case against schema theory (at least as they see it), the authors might have selected, for example, a bimodal distribution, where there is little weight near the mean.

The authors mention pre-registering their results for both experiments. However, scrutinizing their preregistrations suggests that they do not always follow the exact methods they preregistered, which is problematic. They cannot expect every reader to read the preregistration and compare it to what they did. We give two examples below.

* Experiment 1 pre-registration noted it would use BIC values to see how many participant generated sequences from a target distribution (e.g. uniform) would be better fitted by the target distribution or not (uniform vs. Gaussian) and test that against chance. Did this turn into the “Shape” value reported in the manuscript (which dies overall seems like a better measure)? If so, please clarify how. It’s not clear what the reason for the change was.

* A “sequence determinism” based on repeated subsequence exclusion criteria was mentioned in the pre-registration. Please clarify whether this was implemented but no participants were excluded because of it, or if it was not implemented, why?

To their credit, the authors provide a related manuscript, Castillo et al. (2021), to aid with the review process. However, their Experiment 2 was already reported there. So, the new things in this manuscript are Experiment 1 and to the overall discussion. Naturally, this limits the novelty of their current manuscript.

A related issue is that the manuscript is not easy to read and understand in general. In particular, without reading Castillo et al. (2021), it was very difficult to understand what they did in Experiment 2. This puts an unreasonable onus on the reader to read another paper to understand this one. At a minimum, the authors should go over the manuscript carefully and revise it to read more comprehensively. The manuscript should also be self-contained, without having to read other papers and the preregistration to fully understand what it is about.

As noted above, the author’s main effort seems to be to apply their local-sampling approach to randomness, as a sort of “case study” for their theory. Unfortunately, this shows. The authors do not appear to be that familiar with the literature on randomness (see above). Too little of that literature is discussed in the introduction (which might be fine as shorter introductions are now preferred). But they also have a very short discussion, which barely relates their results to the rest of the literature. To give just a few examples, the literature on randomness discusses differences in random-number generation ability following different instructions (Guseva, Bogler, Allefeld, & Haynes, 2023), depending on whether the task involved competition with feedback (Budescu & Rapoport, 1992; Wong, Merholz, & Maoz, 2021), and following many other manipulations. Again, these are just a few examples. Does their model predict any of these findings? Does it have anything to say about Jahanshahi’s model?

Code reproducibility: Seems like the code should work but we could not compile it on our local computer due to missing dependencies. Could authors send a docker container or list what needs to be installed to run their provided code? This seems to be a prerequisite of the journal.

Minor comments

Please clarify whether the participant reshuffled sequences and shape measure used the sequences before or after removing repetitions.

The authors write that they focused on deviations from serial independence, as they are more interpretable than compressibility measures and can easily be applied to non-uniform distributions. This seems like weak justification. First, non-compressibility is one (some might say “the”) measure of randomness. Serial independence is another. They could provide further justification here.

Reviewer #3: The authors study how humans can generate random sequences and find that their sequence generation deviates from iid sampling. They conduct 2 online experiments, one which uses their implicit belief about heights and the other which uses a learned sequence eliminating any prior belief effects. They find a similar deviation from iid sampling across experiments and evidence consistent with earlier studies. Next, they perform a formal model comparison between schema-type models (proposed earlier) and local sampling models. They found support for the local sampling models in experiment 1 and the 2D condition in experiment 2 but not in the 1D condition of experiment 2. However, they found the local sampling models explained certain characteristics of the model better in the 1D condition. Overall on the population level across conditions, they found strong support for the recycled MCMC model.

Overall, I think this paper is well written and tackles a specific problem for which they present empirical insights and perform sophisticated model comparisons. However, I have a few questions that a few questions/comments that I would like the authors to address. I apologize if any of the answers are already present in the text and I missed them. In such a case, highlighting them more would increase the readability of the paper

1) The key confound that worries me regarding the evaluation of different mechanisms is how well the fitted canonical distribution matches the implicit distribution from which the participants draw the random sequences. To address this there are several possible suggestions:

(a)Perform an additional experiment where the participants are either asked to sketch out the 1D distribution after the sample draw or implicitly are asked questions that tell insights about the distribution (quantiles for eg.) to estimate the actual distribution

(b)Without collecting new data, it would be nice to see some QQ plot-type comparisons showing absolute goodness of fits of the fitted and data distribution

(c) Along the lines of the above, plotting the histogram of the CDF of the canonical distribution evaluated at each of the participants' samples would be useful to see. If the canonical distributions are a perfect match, this would be a uniform distribution (CDF of samples drawn from the PDF are uniformly distributed)

2) Did you try a mixture of canonical distributions to see if the quality of fit was better?

3) The results report best-fit Beta distributions or normal distributions. But as I understand, different distributions were fit for each participant. So, is the reported value a mean? Expanding on the clarity here would be helpful.

4) Could you compute and add errorbars to the Fig. 2B (maybe through bootstrapping)

5) How did the sample distributions change over the course of generation? A QQ plot (or equivalent) of the first half and second half would be useful to see

6) A slightly big-picture question is how the explicit random sequence generation compares the implicit samples that participants could generate during Bayesian inference (as in the papers the authors reference). There are problems with explicit questions as choice biases enter the decision process (see https://elifesciences.org/articles/71866 for an example of this). There is no easy way to answer this with the data collected but if new data collection was possible one potential way would be to add a loss function for which the samples are generated (say to estimate the 3 most common heights or so), how the samples differ depending on the loss function. If the samples differed, asking participants to generate random samples as in the current experiment may depend on the implicit loss function they generate the samples for thereby being different for each participant. This is a very broad question so even if new data collection to answer this is not possible, I would be interested to hear the authors’ thoughts on this.

7) It would be useful to see an autocorrelation plot of the participants' samples and those generated by different schemes.

**Have the authors made all data and (if applicable) computational code underlying the findings in their manuscript fully available?**

Reviewer #1: Yes

Reviewer #2: **No: **As discussed in the review, we were unable to run the code as is

Reviewer #3: Yes

PLOS authors have the option to publish the peer review history of their article (what does this mean?). If published, this will include your full peer review and any attached files.

Reviewer #1: **Yes: **C. Philip Beaman

Reviewer #2: No

Reviewer #3: No
---

## [Decision Letter · Decision Letter 1]

22 Nov 2023

Dear Mr Castillo,

Thank you very much for submitting your manuscript "Explaining the Flaws in Human Random Generation as Local Sampling with Momentum" for consideration at PLOS Computational Biology. As with all papers reviewed by the journal, your manuscript was reviewed by members of the editorial board and by several independent reviewers. The reviewers appreciated the attention to an important topic. Based on the reviews, we are likely to accept this manuscript for publication, providing that you modify the manuscript according to the review recommendations.

As you will se below, two of the reviewers still had some concerns. We believe these should be possible to address through a minor revision, but are nevertheless important to address.

Sincerely,

Ulrik R. Beierholm

Academic Editor

PLOS Computational Biology

Daniele Marinazzo

Section Editor

PLOS Computational Biology

Reviewer's Responses to Questions

**Comments to the Authors:**

Reviewer #1: Review by C. Philip Beaman

I previously considered the paper technically sound and I still think the same. The framing and positioning of the paper, and in particular the comparison with Cooper's earlier work which I previously queried, has improved. Inevitably there are still a few things in the presentation I might personally take issue with but I don't see anything here that precludes publication.

Reviewer #2: We thank the authors for taking our comments seriously. We think that the current version of this manuscript is much improved from its previously submitted one. We have only 1 major remaining question.

Major

1. Thank you for clarifying that a participant’s reshuffled sequence did not have repeated items removed. Since this is the case, can the authors further provide an explanation for why they remove repeated items to calculate adjacencies, turning points, and distances for each participant’s sequence, but do not remove repeated items after reshuffling? I am a little confused because at first glance, it seems that a fair comparison would require one to conduct the exact same analysis steps to both the experimental condition (i.e., participant’s sequence) and the control (i.e., the participant’s shuffled sequence). Otherwise, it seems superfluous to make statements such as “Compared to their reshuffled sequences, 208 participants’ values were lower for Repetitions (Observed = .015, Expected = .043, Z = −2.61, p = .009, 209 d = −0.67, BF10 = 4),” (lines 208-209). Is this difference not created by necessity, since repeated items are removed from a participant’s original uttered sequence, but not the reshuffled ones? This point would apply to the repetitions measure, adjacencies measure (line 172-174 mention the removal), turning points measure (lines 181-182), and distance measure (lines 183-184).

Minor

* Line 24: Sentence does not seem grammatical. Did the authors mean to write: "When explicitly asked to be random, people are not random enough."

* Wong et al. (2021) is used as an example that human-attempted random sequences are more compressible than those generated by machines. But that paper demonstrates that humans' randomness is statistically indistinguishable from that of computers in a competitive environment with feedback (when measuring randomness with compressibility).

Reviewer #3: I want to first thank the authors for performing the additional required analyses and addressing my comments. I have one concern that I would like the authors to address:

Given the QQ plot in A3, there seems to be systematic deviation for the empirical distribution for most participants as compared to the best bit underlying distributions. While I understand and accept that a better fit to the theoretical distribution would improve the shape value and provide stronger evidence in favor of the sampling models compared to the schema, it is unclear how these deviations manifest in the deviations from IID sampling and various local sampling models that the authors consider.

For eg. if the underlying distributions the participants were sampling from was flatter than the theoretical distribution (as would be evidenced from the flat part in the QQ plots seen in several participants), then the distances even for IID samplers would be smaller than what is used for comparison. The other metrics would also differ. So to address this:

1) The authors could either using a nonparametric approximation of the empirical distribution (binning finely) as the target distribution or use a mixture distribution to get a better match and perform the subsequent analyses with the better fit target distribution

2) Or if the above analysis is not feasible, then they could subselect the participants whose QQ plot suggest non-significant deviation from the assumed theoretical distribution and compute the measures for those participants (if sufficient in number to draw useful insights).

**Have the authors made all data and (if applicable) computational code underlying the findings in their manuscript fully available?**

Reviewer #1: Yes

Reviewer #2: Yes

Reviewer #3: Yes

PLOS authors have the option to publish the peer review history of their article (what does this mean?). If published, this will include your full peer review and any attached files.

Reviewer #1: **Yes: **C. Philip Beaman

Reviewer #2: No

Reviewer #3: No

Figure Files:

Data Requirements:

Reproducibility:

References:

---

## [Decision Letter · Decision Letter 2]

5 Dec 2023

Dear Mr Castillo,

We are pleased to inform you that your manuscript 'Explaining the Flaws in Human Random Generation as Local Sampling with Momentum' has been provisionally accepted for publication in PLOS Computational Biology.

Best regards,

Ulrik R. Beierholm

Academic Editor

PLOS Computational Biology

Daniele Marinazzo

Section Editor

PLOS Computational Biology

Reviewer's Responses to Questions

**Comments to the Authors:**

Reviewer #2: Thank you for responding to our comments. We have no further major concerns that would preclude publication.

Reviewer #3: Thank you for conducting the additional analysis to rule out the confound and I am happy with the results that still support the conclusions selecting the participants whole responses are approximated well by the theoretical distributions.

**Have the authors made all data and (if applicable) computational code underlying the findings in their manuscript fully available?**

Reviewer #2: Yes

Reviewer #3: Yes

PLOS authors have the option to publish the peer review history of their article (what does this mean?). If published, this will include your full peer review and any attached files.

Reviewer #2: No

Reviewer #3: No

---

## [Editor Report · Acceptance letter]

28 Dec 2023

PCOMPBIOL-D-23-00920R2 

Explaining the Flaws in Human Random Generation as Local Sampling with Momentum

Dear Dr Castillo,

I am pleased to inform you that your manuscript has been formally accepted for publication in PLOS Computational Biology. Your manuscript is now with our production department and you will be notified of the publication date in due course.

With kind regards,

Zsofi Zombor
